# A Sensitive, Point-of-Care Detection of Small Molecules Based on a Portable Barometer: Aflatoxins In Agricultural Products

**DOI:** 10.3390/toxins12030158

**Published:** 2020-03-03

**Authors:** Weiqi Zhang, Wenqin Wu, Chong Cai, Xiaofeng Hu, Hui Li, Yizhen Bai, Zhaowei Zhang, Peiwu Li

**Affiliations:** 1Oil Crops Research Institute of the Chinese Academy of Agricultural Sciences, Wuhan 430062, China; 82101175194@caas.cn (W.Z.); wuwenqin02@gmail.com (W.W.); 82101185206@caas.cn (C.C.); huxiaofeng@caas.cn (X.H.); lihui04@caas.cn (H.L.); baiyizhen@caas.cn (Y.B.); peiwuli@oilcrops.cn (P.L.); 2Key Laboratory of Detection for Mycotoxins, Ministry of Agriculture, Wuhan 430062, China; 3National Reference Laboratory for Agricultural Testing (Biotoxin), Wuhan 430062, China; 4Key Laboratory of Biology and Genetic Improvement of Oil Crops, Ministry of Agriculture, Wuhan 430062, China; 5Laboratory of Risk Assessment for Oilseeds Products, Wuhan, Ministry of Agriculture, Wuhan 430062, China

**Keywords:** aflatoxin, mycotoxin, Au@PtNPs, immunoassay, handheld barometer

## Abstract

Sensitive and point-of-care detection of small toxic molecules plays a key role in food safety. Aflatoxin, a typical small toxic molecule, can cause serious healthcare and economic issues, thereby promoting the development of sensitive and point-of-care detection. Although ELISA is one of the official detection methods, it cannot fill the gap between sensitivity and point-of-care application because it requires a large-scale microplate reader. To employ portable readers in food safety, Pt-catalysis has attracted increasing attention due to its portability and reliability. In this study, we developed a sensitive point-of-care aflatoxin detection (POCAD) method via a portable handheld barometer. We synthesized and characterized Au@PtNPs and Au@PtNPs conjugated with a second antibody (Au@PtNPs-IgG). A competitive immunoassay was established based on the homemade monoclonal antibody against aflatoxins. Au@PtNPs-IgG was used to catalyze the production of O_2_ from H_2_O_2_ in a sealed vessel. The pressure of O_2_ was then recorded by a handheld barometer. The aflatoxin concentration was inversely proportional to the pressure recorded via the barometer reading. After optimization, a limit of detection of 0.03 ng/mL and a linear range from 0.09 to 16.0 ng/mL were achieved. Recovery was recorded as 83.1%–112.0% along with satisfactory results regarding inner- and inter-assay precision (relative standard deviation, RSD < 6.4%). Little cross-reaction was observed. Additionally, the POCAD was validated by high-performance liquid chromatography (HPLC) by using peanut and corn samples. The portable POCAD exhibits strong potential for applications in the on-site detection of small toxic molecules to ensure food safety.

## 1. Introduction

Aflatoxins are mainly synthetized from a polyketide pathway of *Aspergillus flavus* and *Aspergillus parasiticus* [1]. Aflatoxins have been considered as the major food safety threat since their initial detection in the 1960s [2]. The toxicity of aflatoxin is essentially due to: (i) acute aflatoxicosis resulting in hepatic damage, alimentary tract harm, and even death [3]; and (ii) chronic exposure resulting in mutagenic and hepatotoxic effects, immune suppression, and cancer [4]. It was also reported that aflatoxins induce 4.6%–28.2% of global hepatocellular carcinoma [5]. The major agricultural products and food matrices are easily contaminated by aflatoxins from farm to table under favorable environmental conditions, especially in peanut and corn [6]. Serious aflatoxin-induced food safety issues cause billions of dollars to be lost in trading and health care. Currently, many strict regulations of aflatoxin have been set by most regions, such as GB 2761–2017 of China, with the level ranging from 2 to 20 ng/mL [7,8]. Thus, monitoring aflatoxin contamination in agro-food is essential, due to a growing demand for on-site aflatoxin monitoring.

The typical analytical methods for aflatoxin detection are high-performance liquid chromatography (HPLC) and high-performance liquid chromatography-tandem mass spectrometry (HPLC-MS/MS) [8,9]. These approaches offer excellent sensitivity and stability and are high-throughput. Because these methods require extensive time and labor, expensive instruments, and skilled technicians, they cannot satisfy the need for point-of-care aflatoxin detection (POCAD). Enzyme-linked immunosorbent assay (ELISA) boasts the merits of high sensitivity and specificity and easy operation [10]. However, the requirement for a microplate reader has hampered the wide application of ELISA as an on-site detection method.

One serial of point-of-care detection method has been developed by using different signal-amplifying nanomaterials as substitutes for horseradish peroxidase (HRP), such as the introduction of the emerging platinum nanoparticles (PtNPs) [11,12,13]. The catalytic efficiency of PtNPs has been shown to be 400 times higher than that of catalase [14]. Due to the high cost and low availability of PtNPs, monometallic catalysts are substituted by bimetallic or multimetal catalysts [15,16]. Gold nanoparticles (AuNPs) are useful nanostructures due to their great biocompatibility, stability, and optical and electronic properties [15]. The Au@PtNPs, gold core platinum cell nanoparticles, were effectively introduced in catalysis [17], immunochromatographic strips [18], and bar-chart chips [19,20]. To the best of our knowledge, there are few reports on the use of Pt in the detection of small molecules in the context of food safety.

In this paper, we synthesized Au@PtNPs-IgG (connection of Au@PtNPs and goat anti-mouse antibody) as a signal amplifier and then established a sensitive method for point-of-care detection of small molecules based on a handheld barometer with aflatoxins in agro-food as an example. Under optimal parameters, we evaluated the limit of detection (LOD), linear range, and average recovery. This POCAD was further validated by high-performance liquid chromatography (HPLC) using spiked peanut and corn samples. This POCAD can be widely applied in food safety and environmental monitoring.

## 2. Results and Discussion

### 2.1. Principle of POCAD

As reported in Figure 1, the entire POCAD was conducted in a sealed microwell. First, the aflatoxin B_1_ (AFB_1_) antigen was coated on the bottom of the microwell before the extracting sample solution and monoclonal antibody (mAb) against aflatoxins were simultaneously added. After washing steps, the Au@PtNP-IgG was added into the microwell. Later, the rapid addition of H_2_O_2_ produced O_2_ in the sealed microwell. The pressure in the microwell is in inverse proportion to the aflatoxin concentration. By using the handheld barometer, we recorded the pressure in the microwell and calculated the aflatoxin concentration via the external standard method.

### 2.2. Characterization of Au@PtNPs

We employed X-ray diffraction (XRD), UV-vis, high-resolution transmission electron microscopy (HRTEM), and selected area electron diffraction (SAED) for Au@PtNP characterization. In Figure 2A, XRD peaks at 81.82°, 77.48°, 64.62°, 44.44°, and 38.32° correspond to (222), (311), (220), (200), and (111) planes of Au (JCPDS-04-0784) [21]. The other peaks at 85.70°, 81.06°, 67.43°, 46.30°, and 37.76° were in agreement with (222), (311), (220), (200), and (111) planes of Pt (JCPDS-04-0802) [22]. These XRD results demonstrated the successful fabrication of the Au@PtNPs. The Au@PtNPs were characterized using UV-vis spectroscopy (Figure 2B). Compared with the Au peak, the Au@PtNP peak was slightly blueshifted from 521 nm to 517 nm, corresponding to previous work [23]. The HRTEM pattern indicated that the Au@PtNPs (Figure 3A) was homogenous. Comparing with the HRTEM pattern of AuNPs (Figure 3B), we found an increasing diameter from 13 to 16 nm for Au@PtNPs. This finding suggested that a thin layer of Pt was successfully deposited on the AuNPs surface. In addition, the SAED pattern of Au@PtNPs (Figure 3C) exhibited the intense spot of Pt shells in a ring pattern, which matched well with (222), (311), (220), (200), and (111) planes. The SAED results indicated that the crystalline structure of Au@PtNPs was the face-centered-cubic (fcc) crystal structure, which was in good agreement with the XRD result.

### 2.3. Synthesis of Au@PtNPs-IgG

In the synthesis of Au@PtNPs-IgG, we optimized the pH value, blocking buffer and IgG consumption to achieve the optimal POCAD performance. In the optimization of the pH value, we added 1–2 μL K_2_CO_3_ (0.5 M) and observed Au@PtNP coagulation. When we added 3–10 μL K_2_CO_3_ (0.5 M), UV-vis results suggested a similar peak at approximately 523 nm in Figure 4A. The highest optical destiny (OD) produced with an additional 3 μL of K_2_CO_3_ (0.5 M) was chosen as the optimized value with pH at 7.3. The optimized results of the blocking buffer are shown in Figure 4B. Compared with OVA-PBS and Milk-PBS, the use of casein-PBS (w/v) enabled better sensitivity. We further investigated the use of IgG. During the synthesis of Au@PtNPs-IgG, if the IgG final concentration was less than 12 μg/mL, then coagulation was produced: we observed a homogeneous solution for the IgG final concentration above 12 μg/mL. According to the UV-vis results (Figure 4C), we chose the IgG final concentration of 16 μg/mL, which corresponded with the highest OD value. We further conducted POCAD to evaluate the sensitivity with IgG final concentrations ranging from 12–20 μg/mL, resulting in the same final IgG concentration of 16 μg/mL with the best sensitivity (Figure 4D).

### 2.4. Characterization of Au@PtNPs-IgG

As shown in Figure 2B, we observed the obvious redshift from 517 to 531 nm after Au@PtNPs were conjugated with IgG along with the slight enhancement of the optical intensity. The reason for the redshift was the growing diameter of Au@PtNPs before and after conjugation with IgG. The colors of Au@PtNPs and Au@PtNPs-IgG were darker compared with the red color of AuNPs. From the HRTEM analysis, the diameter of Au@PtNPs increased slightly after conjugation with IgG, indicating the successful conjugation of IgG on Au@PtNPs (Figure 3B,D).

The elemental mapping results (Figure 3E–J) proved the positive results of Au@PtNPs-IgG synthesis. As demonstrated through Figure 3E, F and G, the elemental mapping results indicated that the structure of Au@Pt was successfully formed. Comparing Figure 3J with Figure 3E–I, we found that N and O elements were mainly distributed around Au@PtNPs, which indicated that the antibody was successfully connected to Au@PtNPs (the main elements of antibodies are C, H, O, and N. In this paper, we select N and O as value standards because the base of the carrier was a membrane of C).

According to energy dispersive spectroscopy (EDS) pattern results (Figure 5), we observed the apparent growing intensity of C, N, and O, suggesting the conjugation of IgG with Au@PtNPs. Specifically, other than Au@PtNPs, N only appeared in Au@PtNPs-IgG. The intensity values of C and O increased 78.9% and 509%, respectively.

### 2.5. Optimization of POCAD

#### 2.5.1. Blocking Solution for Immunoassay

The blocking buffer for the immunoassay would influence the immunoassay results. Three kinds of blocking buffer were studied: 1.5% OVA-PBST (w/v), 0.25% casein-PBST (w/v), and 2.5% nonfat milk-PBST (w/v). As shown in Figure 6A, better sensitivity was found when we employed 2.5% milk-PBST.

#### 2.5.2. Methanol Concentration for Immunoassay

The concentration of methanol has a significant effect on the combination of antibody and antigen. The sensitivity of this system decreases with increasing amounts of methanol. In this research, we have compared a series of concentrations of methanol including 0%, 10%, 20%, and 40%. We determined that 10% methanol was better than the other concentrations due to its superior sensitivity, shown in Figure 6B. After the optimization of these items above, we could obtain an excellent parameter combination.

### 2.6. Standard Curve and LOD

By using the aflatoxins standard solution, we obtained the standard curve (Figure 7) under the optimal situation. Through nonlinear fitting, we obtained Formula (1) to calculate the content of aflatoxins in detected samples. By calculating, we obtained the half-maximal inhibitory concentration (IC_50_) of POCAD of 0.24 ng/mL with the LOD of 0.03 ng/mL. The linear range of this approach was 0.09–16.0 ng/mL.
y = 0.45 + 0.55/(1 + (x/0.24)^1.12^), R^2^ = 0.997(1)

The sensitivity comparison between POCAD and the previous study are reported in Table 1. According to the date reported in the previous works, the sensitivity of POCAD was at least three times lower than the other methods.

### 2.7. Recovery

To validate the accuracy of POCAD, we employed this method to detect negative samples (peanut) with additional aflatoxins. From this detection, we could obtain the average recovery by the pressure readout method. The aflatoxins in detected samples were quantitated through the standard curve. As shown in Table 2, including inner-assay and inter-assay results, the strong average recovery of this method (83.1%–112.0%) indicated that this method was satisfactory for spiked sample detection.

### 2.8. Within Assay and between Assay

By analyzing the inner-assay and inter-assay in Table 2, we could calculate that the relative standard distribution (RSD) values separately ranged from 2.7% to 5.8% and 2.3% to 6.4%. These results indicated that POCAD was stable and suitable for development as a rapid aflatoxin detection method.

### 2.9. Validation via HPLC Using Peanut and Corn Samples

To estimate the accuracy of POCAD, we employed the six agricultural samples (peanut and corn) to perform this method and HPLC separately. The sample preparation and method performance were conducted according to the protocol described above. By comparing the results of POCAD and HPLC separately (shown in Table 3), we found proper agreement between the two analytical methods, as presented in the results (y = 0.22 x + 0.99, R^2^ = 99.2%).

## 3. Conclusions

The point-of-care detection of hazardous small molecules is important in agro-food safety. Using aflatoxin as an example, we developed a sensitive point-of-care detection method for small molecules based on a portable handheld barometer. The Au@PtNPs were synthesized and then conjugated with IgG as a signal amplifier. Based on the excellent catalytic efficiency of Au@PtNPs, the immuno-reaction results indicated effective O_2_ generation, which produced increased pressure signals in a sealed system. By using a handheld gas meter, we obtained the aflatoxin concentration rapidly and precisely. After optimization, we recorded the LOD of 0.03 ng/mL, and a linear range of 0.09 to 16.0 ng/mL along with average recovery of 83.1%–112.0% and slight cross-reactivity. The inner-assay and inter-assay results demonstrated satisfactory RSD (<6.4%). Accuracy was further validated by a typical HPLC method. This proposal suggests potential application in POC detection in food safety.

## 4. Materials and Methods

### 4.1. Chemicals and Instruments

Aflatoxins mix standard solution, 4-(2-hydroxyethyl)-1-piperazineethanesulfonic acid (HEPES), HAuCl_4_·3H_2_O, and H_2_PtCl_6_·xH2O, 30% H_2_O_2_ were obtained from Sigma-Aldrich, Inc. (Shanghai, China). Trisodium citrate, ascorbic acid, Tween 20, and sucrose were obtained from Sinopharm Chemical Reagent Co., Ltd. (Beijing, China). Phosphate-buffered saline (PBS, 0.01 M, pH 7.4) was prepared by adding 2.9 g of Na_2_HPO_4_·12H_2_O, 0.2 g of KCl, 0.2 g of KH_2_PO_4_, and 8 g of NaCl into 1 L of deionized water. Carbonate buffer (0.05 M, pH 9.6) was prepared by adding 2.93 g of NaHCO_3_ and 1.59 g of Na_2_CO_3_ into 1 L of deionized water. Water was purified by a MilliQ system (Millipore, Danvers, MA, USA). All reagents applied were of analytical grade unless otherwise specified. All of the glassware was cleaned with aqua regia (HNO_3_:HCl = 1:3(v/v)) before use. The monoclonal antibody (mAb) 1C11 for aflatoxins was produced by our own laboratory [28]. Goat anti-mouse antibody was obtained from GE Healthcare Co., Ltd. (Piscataway, NJ, USA).

The ultraviolet spectrum was acquired by a SpectraMax M2e microplate reader (Molecular Devices Corp., Sunnyvale, CA, USA). X-ray diffraction (XRD) patterns were obtained by a Bruker D8 Advance instrument (Bruker AXS Co. Ltd., Karlsruhe, Germany) with a Cu Ka radiation source (λ = 0.15406 nm). High-resolution transmission electron microscopy (HRTEM), elemental mapping and energy dispersive spectroscopy (EDS) were performed using a JEM-2100F TEM facility (Jeol Ltd., Tokyo, Japan).

### 4.2. Synthesis and Characterization of Au@PtNPs-IgG

First, AuNPs were synthesized by a reduction method [29]. Four milliliters of 1% trisodium citrate was added into 200 mL of 0.01% (w/v) HAuCl_4_ under reflux. The reaction was refluxed for 20 min before cooling to room temperature naturally, and the AuNPs were stored at 4 °C before use. Second, the Au@PtNPs were synthesized via seed-mediated growth. Ten milliliters of AuNPs solution was mixed with 200 μL of 3.86 mM H_2_PtCl_4_ at 80 °C with gentle stirring for 5 min. Then, 400 μL of 10 mM ascorbic acid was added dropwise and kept at 80 °C for 30 min. After cooling naturally, the Au@PtNPs were stored at 4 °C until further use.

Before the synthesis of Au@PtNPs-IgG, the pH value of Au@PtNPs was optimized. After 1–10 μL K_2_CO_3_ (0.5 M) was added into 1 mL of Au@PtNPs solution to adjust the pH value, the optimal pH of the reaction solution was confirmed through UV-vis. After that step, a saturated concentration of anti-mouse antibody was added into Au@PtNPs solution and incubated for 30 min at 37 °C. Then, the equal volume of blocking buffer (0.25% OVA-PBS, 0.25% milk-PBS, 0.25% casein-PBS, w/v) was added and incubated at 37 °C for 1 h. After centrifugation (9000 g at 4 °C for 10 min), the precipitate was resuspended with the stock solution (10 mM HEPES, 10 mM citric acid, 0.1% Tween 20, 5% sucrose, pH = 7). The Au@PtNPs-IgG solution was stored at 4 °C before use.

We synthesized the Au@PtNPs-IgG solution by changing the amount of the IgG (final concentration: 10–20 μg/mL). The optimal amount of IgG was confirmed through UV-vis spectrum scanning from 400–700 nm.

UV-visual spectroscopy was employed to characterize AuNPs, Au@PtNPs, and Au@PtNPs-IgG. The crystalline structure of Au@PtNPs was identified by X-ray diffraction with angles from 30° to 90°, as well as selected area electron diffraction (SAED). The pattern of EDS exhibited all elements of Au@PtNPs. The morphological characteristics of AuNPs, Au@PtNPs, and Au@PtNPs-IgG were analyzed by high-resolution transmission electron microscopy (HRTEM). The elemental mapping of the pattern of the energy dispersive spectrometer exhibited the elements of Au@PtNPs-IgG. The elemental mapping showed a variety of elements and positions of Au@PtNPs-IgG.

### 4.3. Fabrication of POCAD

The POCAD comprised a microwell plate and a handheld barometer. A rubber strip was used to seal the microwell plate as a cover. After POCAD was conducted, a portable handheld barometer was used to measure the gas pressure in the sealed microwell.

### 4.4. Procedure of POCAD

One hundred microliters of 0.05 M carbonate buffer containing AFB_1_-BSA (0.5 μg/mL) in a microwell plate was incubated at 4 °C for 12 h. After washing three times with 0.05% PBST (0.01 M PBS with 0.05% Tween 20), the well was blocked with a blocking buffer at 37 °C for 1 h. The microwell was washed three times with 0.05% PBST. In a microwell, 100 μL of the mixture of aflatoxin standard solution (final concentration ranged from 0–20 ng/mL in 5% MeOH) and monoclonal antibody (final concentration: 0.5 μg/mL) was incubated at 37 °C for 1 h and then washed three times with 0.05% PBST. One hundred microliters of Au@PtNPs-IgG solution was added into these wells and incubated at 37 °C for another 1 h. The microwell was washed six times with 0.05% PBST to ensure that superfluous Au@PtNPs-IgG was removed. Then, 100 μL of H_2_O_2_ was added. The catalysis of Au@PtNPs-IgG on H_2_O_2_ was conducted for 15 min. The pressure values of the microwell were obtained by a handheld barometer.

### 4.5. Optimization of POCAD

We analyzed 1.5% OVA-PBST (w/v), 0.25% casein-PBST (w/v), and 2.5% nonfat milk-PBST (w/v) as blocking buffers. The concentration of methanol was also optimized with final concentrations of 0%, 5%, 10%, and 20%. The best blocking buffer and the optimal final methanol concentration were recorded according to the sensitivity of POCAD.

### 4.6. Evaluation of POCAD

#### 4.6.1. LOD and Linear Range

The POCAD standard curve was determined via a series of aflatoxin standard solutions (0–10 ng/mL). The standard curve was established by fitting the inhibitory concentration value versus the concentration of aflatoxins using four parameter logistic regression. The LOD, sensitivity, and linear range were calculated according to previous research [30].

#### 4.6.2. Recovery, Inner-Assay Precision and Inter-Assay Precision

To evaluate the recovery, inner-assay precision, and inter-assay precision of POCAD, we spiked aflatoxins (final concentration: 5–20 ng/mL) in blank peanut and corn samples. Then, the samples were ground into homogenized powders and stored overnight at room temperature. Under optimization, we used the POCAD to record the recovery.

The inner-assay experiment was conducted by detecting spiked samples (final concentration: 5–20 ng/mL) of aflatoxins within one day, while the inter-assay experiment was conducted by using the spiked samples (final concentration: 5–10 ng/mL) on day 1 and day 5.

### 4.7. Validation via Immune Affinity Column-HPLC

The POCAD was validated with the immune affinity column (IAC)-HPLC-FLD (fluorescence detector) method. We used homemade IAC with 10 mL dilution, as in our previous protocol [31]. In HPLC, we employed a C_18_ column (particle size 3 μm, 150 mm × 4.6 mm). The excitation wavelength of the fluorescence detector was set as 360 nm, while the emission wavelength was set as 440 nm. The injection volume of samples was fixed as 10 μL, while the column temperature was confirmed as 30 °C The mobile phase was 45% methanol–water solution at the flow rate of 1 mL/min.

### 4.8. Sample Pretreatment

Peanut and corn samples, purchased from a local supermarket, were used to evaluate the performance of the POCAD method. After grinding, the homogenized peanut and corn powder samples (5 g) were added to 20 mL of 80% (v/v) methanol solution and homogenized with a high-speed vortex mixer (Vortex-Genie2, SI, New York, NY, USA) for 3 min. After filtration with a glass fiber filter, 10 mL of the filtrate was diluted with four volumes of deionized water. The dilution was stored at 4 °C until further use.

## Figures and Tables

**Figure 1 toxins-12-00158-f001:**
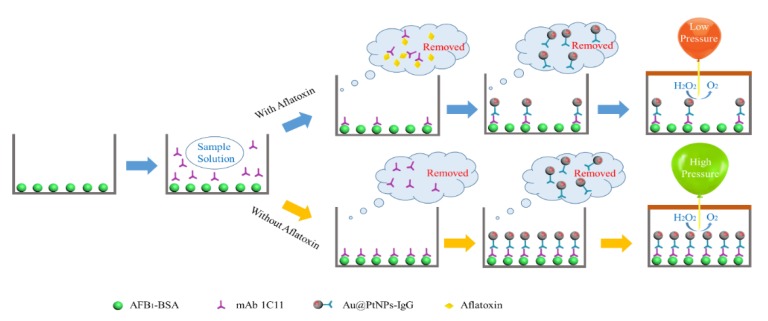
Scheme of point-of-care aflatoxin detection (POCAD) of aflatoxins. The entire POCAD was conducted in a sealed microwell. The AFB_1_ antigen was coated on the bottom of the microwell. During the immunoassay, the aflatoxin and antigen compete with the binding site of the mAb. After adding of Au@PtNP-IgG, Au@PtNP-IgG bind with mAb connected to the antigen. Later, the rapid addition of H_2_O_2_ produced O_2_ in the sealed microwell. The pressure in the microwell is in inverse proportion to the aflatoxin concentration. By using the handheld barometer, we recorded the pressure in the microwell and calculated the aflatoxin concentration via the external standard method.

**Figure 2 toxins-12-00158-f002:**
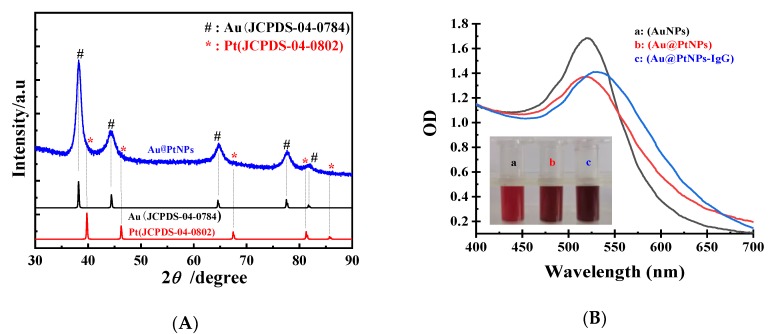
(**A**) X-ray diffraction pattern of Au@PtNPs (black line: Au (JCPDS-04-0784) standard PDF card, red line: Pt (JCPDS-04-0802) standard PDF card, blue line: the diffraction pattern of Au @ PtNPs); (**B**) exhibits UV-visible spectroscopy of AuNPs (a), Au@PtNPs (b), and Au@PtNPs-IgG (c).

**Figure 3 toxins-12-00158-f003:**
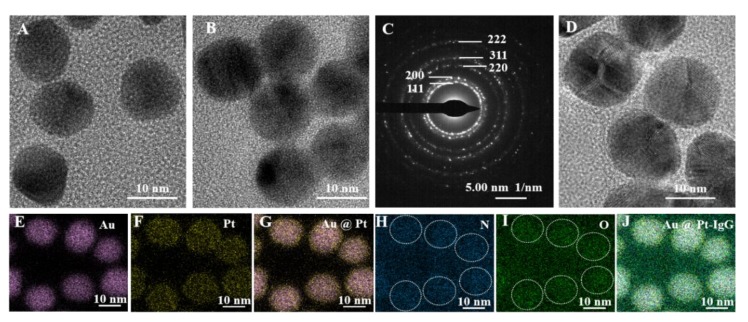
High-resolution transmission electron microscopy (HRTEM) images of Au (**A**), Au@PtNPs (**B**), and Au@PtNPs-IgG (**D**); selected area electron diffraction (SAED) of Au@PtNPs (**C**); and elemental mappings of Au (**E**), Pt (**F**), Au@PtNPs (**G**), N (**H**), O (**I**), and all elements (**J**) in Au@PtNPs-IgG. Scale bar: 10 nm.

**Figure 4 toxins-12-00158-f004:**
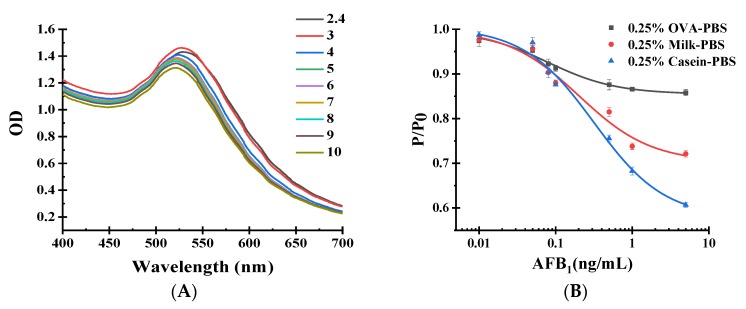
Optimization of Au@PtNPs and Au@PtNPs-IgG: (**A**) UV-visible spectroscopy scanning of Au@PtNPs with 1–10 μL K_2_CO_3_ (0.5 M) to adjust the pH value of the reaction system; (**B**) different blocking buffers for Au@PtNPs; (**C**) UV-visible spectroscopy scanning of Au@PtNPs-IgG among different concentrations (12, 14, 16, 18, 20 μg/mL) of goat anti-mouse antibody; (**D**) different final concentrations (12, 14, 16, 18, 20 μg/mL) of goat anti-mouse antibody in the reaction system.

**Figure 5 toxins-12-00158-f005:**
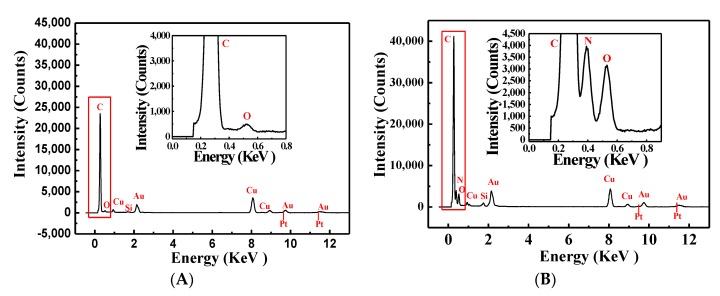
Energy dispersive spectroscopy patterns of Au@PtNPs (**A**) and Au@PtNPs-IgG (**B**), illustrations are the magnification of the corresponding C, N, O element.

**Figure 6 toxins-12-00158-f006:**
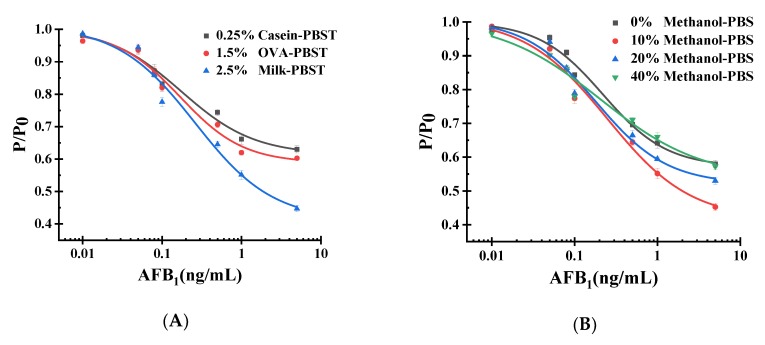
Optimization of the characteristics which influence the pressure results of the reaction system: (**A**) influence of the different blocking buffers for pressure reaction; (**B**) influence of the different concentrations of methanol on the immune reaction.

**Figure 7 toxins-12-00158-f007:**
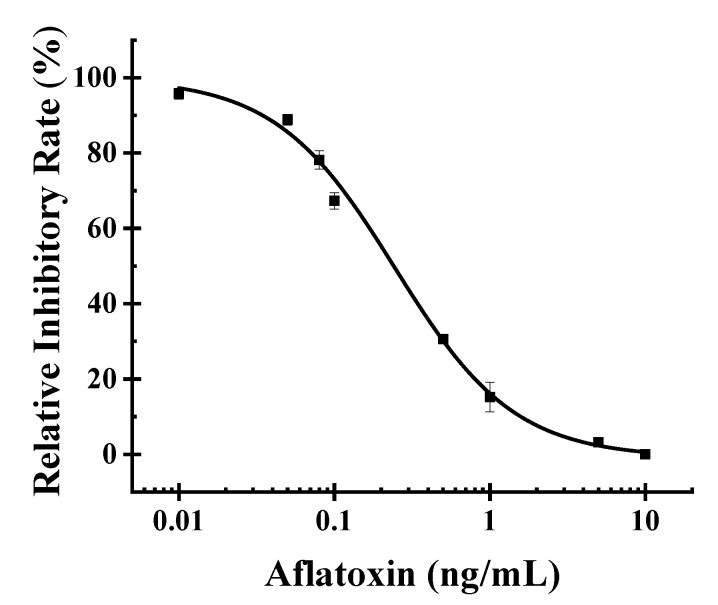
Standard curve for aflatoxin quantitation. The pressure data was obtained by using the aflatoxins standard solution ranging from 0–10 ng/mL. The relative inhibitory rate (RIR) was calculated by the formula (RIR = (1 − (P − P_m_)/(P_0_ − P_m_)) × 100%), where P_0_ and P_m_ were the pressure values corresponding to aflatoxin concentration of 0 and 10 ng/mL, respectively, and P represents the pressure values of other aflatoxin concentrations. The standard curve fit through the formula (y = 0.45 + 0.55/(1 + (x/0.24)^1.12^), R^2^ = 0.997).

**Table 1 toxins-12-00158-t001:** Sensitivity comparison between POCAD and previous reports.

Assay Methods	Assay Target	LOD (ng/mL)	Reference
Europium nanospheres-based time-resolved fluorescence immunoassay	AFTs^1^	0.16	[24]
Fluorescent microspheres-based test strip	AFB_1_	2.5	[25]
Graphene oxide and carboxylated graphene oxide-based test strip	AFB_1_	0.3	[26]
Aptamer based test strip	AFB_1_	0.1	[27]
POCAD	AFTs	0.03	This work

^1^ AFTs is the abbreviation of total aflatoxins.

**Table 2 toxins-12-00158-t002:** Results of recovery analysis by POCAD.

	Spike Level (μg/kg)	Mean ± SD	Average Recovery (%)	Relative Standard Deviation (RSD%)
Within assay (*n* = 3)^a^	5	4.25 ± 0.25^c^	85.0	5.8
	15	16.79 ± 0.72	112.0	4.3
	20	19.33 ± 0.52	96.7	2.7
Between assay (*n* = 5)^b^	5	4.15 ± 0.16	83.1	3.8
	15	14.87 ± 0.34	99.1	2.3
	20	17.68 ± 1.13	88.4	6.4

^a^ The experiments were carried out in three replicates on the same day; ^b^ the assays were carried out on five different days. ^c^ The data are average values and standard deviation (SD).

**Table 3 toxins-12-00158-t003:** Comparison of real samples detection by POCAD and HPLC.

Samples	This Work (μg/kg *n* = 3)^a^	HPLC (μg/kg *n* = 3)
1	4.86 ±0.34^b^	4.87 ± 0.08
2	15.10 ± 0.50	15.04 ± 0.40
3	22.51 ± 0.95	21.64 ± 0.25
4	4.22 ± 0.26	4.13 ± 0.10
5	13.38 ± 0.52	14.01 ± 0.17
6	18.98 ± 0.95	19.62 ± 0.2

^a^ The experiments were carried out in three replicates on the same day. ^b^ The data are average values and SD.

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
