# Peer review of "A Sensitive, Point-of-Care Detection of Small Molecules Based on a Portable Barometer: Aflatoxins In Agricultural Products"

_toxins, 2020, doi:10.3390/toxins12030158_

Round 1
Reviewer 1 Report
The manuscript obtained for assessment concerns a very important issue, which are sensitive methods for determining one of the most dangerous mycotoxins, which are aflatoxins in food. In general, the manuscript is properly prepared but requires some corrections. I place all comments below.
L38 - in "ng / mL"? Please provide the source of legal regulations. They are not listed.
Chapter 2.7 The text mentions the determination of recovery for nuts and maize. However, in Table 2 which values are for which matrices? The authors state that the developed method has a low detection limit. What was the value of the limit of quantification? In a validation experiment recovery at a fairly high level (5 and more µg / kg) was tested. Taking into account European regulations, the acceptable level of AFB1 content in nuts intended for direct consumption is 2µg / kg. I believe that the recovery is for too high a concentration. In addition, the enhancement concerned AFB1 or the sum of all four aflatoxins? How does the linearity range (determined in µg / ml) translate into real concentrations in µg / kg?
Figure 7 - What was aflatoxin?
Author Response
Response to Reviewer 1 Comments
First of all, thanks for your kind and effective work. On behalf of my co-authors, thank you very much for giving us an opportunity to revise our manuscript, we appreciate you very much for the positive and constructive comments and suggestions on our manuscript. The response to your comments and advice were listed as follows:
Point 1: Line 38 - in "ng / mL"? Please provide the source of legal regulations. They are not listed
Response 1: Thank you very much for your detailed work, we have embedded some legal regulations and other references in this article. The sentence of “Currently, many countries set strict regulation levels of aflatoxin, ranging from 2 to 20 ng/mL [7].” has been changed as follows: “Serious aflatoxin-induced food safety issues cause billions of dollars to be lost in trading and health care. Currently, many strict regulations of aflatoxin were set by most regions, such as GB 2761-2017 of China, the level was ranging from 2 to 20 ng/mL [7,8].”.
[7] Xu, L.; Zhang, H.; Yan, X.; Peng, H.; Wang, Z.; Zhang, Q.; Li, P.; Zhang, Z.; Le, X. C. Binding-Induced DNA Dissociation Assay for Small Molecules: Sensing Aflatoxin B1. ACS Sens. 2018, 3, 2590-2596
[8] Yu, L.; Ma, F.; Ding, X. X.; Wang, H. L.; Li, P. W. Silica/Graphene Oxide Nanocomposites: Potential Adsorbents for Solid Phase Extraction of Trace Aflatoxins in Cereal Crops Coupled with High Performance Liquid Chromatography. Food Chem. 2018, 245, 1018-1024.
Point 2: Chapter 2.7 The text mentions the determination of recovery for nuts and maize. However, in Table 2 which values are for which matrices?
Response 2: Thank you very much for your detailed work. In Chapter 2.7, we just employed peanut as detection matrices. And we use peanut and corn samples to estimate the accuracy of POCAD in Chapter 2.9. It was a writing mistake. we have corrected this mistake and I apologize for this mistake, sincerely.
Point 3: The authors state that the developed method has a low detection limit. What was the value of the limit of quantification?
Response 3: Thank you very much for your kind work. The quantification of this method was calculated as 0.09 ng/mL. The data of this method was fitted by the four-parameter logic equation as ELISA curve. The limit of aflatoxin quantification in this method could obtain as the calculation method of ELISA curve using the IC20.
Point 4: In a validation experiment recovery at a fairly high level (5 and more µg/kg) was tested. Taking into account European regulations, the acceptable level of AFB1 content in nuts intended for direct consumption is 2µg/kg. I believe that the recovery is for too high a concentration. In addition, the enhancement concerned AFB1 or the sum of all four aflatoxins?
Response 4: Thank you very much for your kind work. At the beginning of the work to analyze the recovery, we just consider the legislation of aflatoxin limit in China (GB 2761-2017). In that legal document, the limit of aflatoxin in nut ranges from 5.0-20 ng/mL. It is deficient to not take other regulations into account. We promise we will make up this deficiency in future work. In addition, the enhancement concerned the sum of all four aflatoxins.
Point 5: How does the linearity range (determined in µg/ml) translate into real concentrations in µg/kg?
Response 5: Thank you very much for your kind work. We translate the unit (µg/ml) into another unit (µg/kg) by the following formula:
((µg/mL)×V0 )/m0=µg/k
V0 is the volume of sample used in detection;
m0 is the mass of the real sample
At last, allow me to thank you again for your careful advice for our manuscript!

Reviewer 2 Report
The manuscript "A sensitive, point-of-care detection of small 2 molecules based on a portable barometer: aflatoxins 3 in agricultural products" describe adequately the POCAD method for aflatoxin detection.
I have just a question. Have the authors tryed the POCAD method on other aflatoxins?
I report my comments in the attached file

Author Response
Response to Reviewer 2 Comments
First of all, thanks for your kind and effective work. On behalf of my co-authors, thank you very much for giving us an opportunity to revise our manuscript, we appreciate you very much for the positive and constructive comments and suggestions on our manuscript. The response to your comments and advice were listed as follows:
Point 1: Have the authors tried the POCAD method on other aflatoxins?
Response 1: Yes, there are some other students in our group are trying to apply the POCAD method to detect other mycotoxins, such as ochratoxin, Zearalenone.
Point 2: The informal expression of subscript of the entire manuscript.
Response 2: Thank you very much for your detailed work in the subscript correction. We have checked this problem of the entire article and have corrected all the mistakes. The informal expression in original edition were listed as follows: “O2” in line 15, 68, 75, 199; “H2O2” in line 75, 209, 264, 265; “HAuCl4·3H2O” in line 208; “H2PtCl6·xH2O” in line 209; “Na2HPO4·12H2O”, “KH2PO4” in line 212; “NaHCO3” in line 213; “Na2CO3” in line 214; “HAuCl4” in line 227; “H2PtCl4” in line 230; “K2CO3” in line 102, 104, 115, 234.
Point 3: Typeface correction and expression improvement from line 29 to 39.
Response 3: Response: Thank you very much for your kind advice on expression details. We have improved expression of the paragraph and underlined what we have changed as follows:
Original edition: Aflatoxins are mainly metabolized from a polyketide pathway of Aspergillus flavus and Aspergillus parasiticus [1]. Aflatoxins have been regarded as the major food safety threat after their initial detection in the 1960s [2]. The toxicity of aflatoxin relies on two aspects. Acute aflatoxicosis results in hepatic damage, alimentary tract harm, and even death [3]. Chronic exposure results in mutagenic and hepatotoxic effects, immune suppression, and cancer [4]. In total, aflatoxins induce 4.6%-28.2% of global hepatocellular carcinoma [5]. The major agricultural products and food matrices are easily contaminated by aflatoxins from farm to table under certain temperature and humidity conditions, especially in peanut and corn [6].
New edition: Aflatoxins are mainly synthetized from a polyketide pathway of Aspergillus flavus and Aspergillus parasiticus [1]. Aflatoxins have been regarded as the major food safety threat since their initial detection in the 1960s [2]. The toxicity of aflatoxin is due essentially to: i) acute aflatoxicosis resulting in hepatic damage, alimentary tract harm, and even death [3]; ii) chronic exposure resulting in mutagenic and hepatotoxic effects, immune suppression, and cancer [4]. It was also reported that aflatoxins induce 4.6%-28.2% of global hepatocellular carcinoma [5]. The major agricultural products and food matrices are easily contaminated by aflatoxins from farm to table under favourable environmental conditions, especially in peanut and corn [6].
Point 4: This sentence is not clear for the readers. If possible, please, rephrase it, for improving the comprehension. (line 48 to 50)
Response 4: Thank you for your kind advice, we have rephrased the sentence and contrasted as follows.
Original edition: One point-of-care detection method is the development of signal amplifying nanomaterials as substitutes of horseradish peroxidase (HRP), such as the introduction of the emerging platinum nanoparticles (PtNPs) [11-13].
New edition: One serial of point-of-care detection method has been developed by using different signal amplifying nanomaterials as substitutes of horseradish peroxidase (HRP), such as the introduction of the emerging platinum nanoparticles (PtNPs) [11-13].
Point 5: Expression improvement and a conference labeling mistake from line 52 to 54.
Response 5: Thank you for your kind advice, we have rephrased the sentence and corrected the label as follows:
Original edition: Gold nanoparticles (AuNPs) are useful nanostructures because of their great biocompatibility, stability, and optical and electronic properties 15.
New edition: Gold nanoparticles (AuNPs) are useful nanostructures due to their great biocompatibility, stability, and optical and electronic properties [15].
Point 6: What does “Au@PtNPs” mean? Specify. (line 54).
Response 6: Thank you for your detailed advice. We have embedded the detail in the sentence to explain it as follows: “Au@PtNPs, gold core platinum cell nanoparticles, were effectively introduced in catalysis”.
Point 7: What does “Au@PtNPs-IgG” mean? Specify. (line 57)
Response 7: Thank you for your detailed advice. We have embedded the detail in the sentence to explain it as follows: “In this paper, we synthesized Au@PtNPs-IgG (connection of Au@PtNPs and goat anti-mouse antibody) as a signal amplifier”.
Point 8: Expression improvement in line 65.
Response 8: Thank you for your kind advice, we have rephrased the sentence and corrected the label as follows:
Original edition: As seen in Figure 1, the entire POCAD was conducted in a sealed microwell.
New edition: As reported in Figure 1, the entire POCAD was conducted in a sealed microwell.
Point 9: What does “mAb” means? (line 66).
Response 9: Thank you for your detailed work. We have added the explanation about it in the sentence as follows: “monoclonal antibody (mAb)”.
Point 10: Maybe is better describe the details of the method.
Response 10: Thank you for your detailed advice. The calculation method of aflatoxin was the main part of “2.6 standard curve and LOD”, so that we just want to leave it to describe detailly in that part.
The detail 1 was from line 156 to 160: “By using the aflatoxins standard solution, we obtained the standard curve (Figure 7) under the optimal situation. Through nonlinear fitting, we obtained the formula (1) to calculate the content of aflatoxins in detected samples. By calculating, we obtained the half-maximal inhibitory concentration (IC50) of POCAD of 0.24 ng/mL with the LOD of 0.03 ng/mL. The linear range of this approach was 0.09-16.0 ng/mL.”
The detail 2 was from line 164 to 169: “Standard curve for aflatoxin quantitation. The pressure data were obtained by using the aflatoxins standard solution ranging from 0-10 ng/mL. The relative inhibitory rate (RIR) was calculated by the formula (RIR=(1-(P-Pm)/(P0-Pm))×100%), P0 and Pm were the pressure values corresponding to aflatoxin concentration of 0 and 10 ng/mL, respectively, P represent the pressure values of other aflatoxin concentrations). The standard curve was fitting through the formula”.
Point 11: Deletion of needless word “the” in line 114 and 153
Response 11: Thank you for your detailed advice. Both needless words were deleted.
Original edition 1: (a) the UV-visible spectroscopy scanning of Au@PtNPs with 1-10 μL K2CO3 (0.5 M) to adjust the pH value of the reaction system;
Original edition 2: (b) the influence of the different concentrations of methanol on the immune reaction.
Point 12: Insertion of the phrase of “influence of” in line 153
Response 12: Thank you for your detailed advice. We have corrected this mistake.
New edition: (a) influence of the different blocking buffers for pressure reaction;
Point 13: Two numbers after comma are enough. (line 160 and 169)
Response 13: Thank you for your advice about the formula in line 160 and 169.
We have change “y=0.44818+0.55235/((1+((x/0.23662))1.11639)) R2=0.997” to “y=0.45+0.55/((1+((x/0.24))1.12)), R2=0.997”.
Point 14: Expression improvement from 161 to 163
Response 14: Thank you for your advice about the sentence from line 161 to 163. We have rephrased it as follows
Original edition: The sensitivity comparison between POCAD and the previous study was concluded in Table 1. From this table, we could reach the conclusion that the sensitivity of POCAD was at least three times lower than in previous work
New edition: The sensitivity comparison between POCAD and the previous study are reported in Table 1. According to the date reported in the previous works, the sensitivity of POCAD was at least three times lower than the other methods.
Point 15: Expression improvement in 164.
Response 15: Thank you for your kind advice, we have rephrased the sentence as follows:
Original edition: Figure 7. Standard curve to quantitate aflatoxin.
New edition: Figure 7. Standard curve for aflatoxin quantitation.
Point 16: specify that the data are the average value and...what? (the data of Table 3, in line 193)
Response 16: Thank you for your detailed advice. We have supplied an explanation after Table 2 and Table 3 as “the data was average value and standard deviation (SD)” and “the data was average value and SD”.
Point 17: Informal expression in line 212 and 214
Response 17: Thank you for your kind advice, we have rephrased the sentence as follows:
Original edition: 8 g of NaCl into 1,000 mL of deionized water. (line 212)
Carbonate buffer (0.05 M, pH 9.6) was prepared by adding 2.93 g of NaHCO3 and 1.59 g of Na2CO3 into 1,000 mL of deionized water. (line 214)
New edition: 8 g of NaCl into 1 L of deionized water.
Carbonate buffer (0.05 M, pH 9.6) was prepared by adding 2.93 g of NaHCO3 and 1.59 g of Na2CO3 into 1L of deionized water. (line 214)
Point 18: Construction changing of chapter 4.5 (line 268-275)
Response 18: Thank you for your advice about the construction changing of the article 4.5.1. Blocking buffer and 4.5.2 Methanol concentration. We hve rphrased them as follows:
Original edition:
4.5. Optimization of POCAD
4.5.1. Blocking buffer
We chose 1.5% OVA-PBST (w/v), 0.25% casein-PBST (w/v), and 2.5% nonfat milk-PBST (w/v) as blocking buffers for POCAD. Then, we estimated the influences of the 3 blocking buffers on POCAD. According to the sensitivity of POCAD, the optimal blocking buffer was obtained.
4.5.2. Methanol concentration
The concentration of methanol was optimized with final concentrations of 0%, 5%, 10%, and 20%. By comparison with the sensitivity of POCAD, the optimal final methanol concentration was recorded.
New edition:
4.5. Optimization of POCAD
We analyzed 1.5% OVA-PBST (w/v), 0.25% casein-PBST (w/v), and 2.5% nonfat milk-PBST (w/v) as blocking buffers. The concentration of methanol was also optimized with final concentrations of 0%, 5%, 10%, and 20%. The best blocking buffer and the optimal final methanol concentration were recorded according to the sensitivity of POCAD.
Point 19: Redundant description in line 278
Response 19: Thank you for your kind work, we have accepted the deletion of redundant description in line 278 “With the optimal blocking buffer and methanol concentration”.
At last, allow me to thank you again for your careful advice for our manuscript!

Round 2
Reviewer 1 Report
The authors added explanations and corrected the manuscript according to the comments of the reviewers.
Author Response
Dear Reviewer
Thanks for your kind and effective work. On behalf of my co-authors, thank you very much for giving us an opportunity to revise our manuscript, we appreciate you for the positive and constructive comments and suggestions on our manuscript entitled “A sensitive, point-of-care detection of small molecules based on a portable barometer: aflatoxins in agricultural products” (toxins-726990).
